# Stimulus Complexity Can Enhance Art Appreciation: Phenomenological and Psychophysiological Evidence for the Pleasure-Interest Model of Aesthetic Liking

**DOI:** 10.3390/jintelligence12040042

**Published:** 2024-04-03

**Authors:** Tammy-Ann Husselman, Edson Filho, Luca W. Zugic, Emma Threadgold, Linden J. Ball

**Affiliations:** 1School of Psychology & Neuroscience, University of Glasgow, 62 Hillhead Street, Glasgow G12 8QB, UK; thusselman77@gmail.com; 2Wheelock College of Education & Human Development, Boston University, 2 Silber Way, Boston, MA 02215, USA; efilho@bu.edu; 3School of Psychology & Humanities, University of Central Lancashire, Fylde Road, Preston PR1 8TY, UKethreadgold1@uclan.ac.uk (E.T.)

**Keywords:** stimulus complexity, processing fluency, perceptual fluency, conceptual fluency, art appreciation, aesthetic liking, flow-feeling, dual-process theory, electroencephalography (EEG), alpha power

## Abstract

We tested predictions deriving from the “Pleasure-Interest Model of Aesthetic Liking” (PIA Model), whereby aesthetic preferences arise from two fluency-based processes: an initial automatic, *percept-driven* default process and a subsequent *perceiver-driven* reflective process. One key trigger for reflective processing is stimulus complexity. Moreover, if meaning can be derived from such complexity, then this can engender increased *interest* and elevated liking. Experiment 1 involved graffiti street-art images, pre-normed to elicit low, moderate and high levels of interest. Subjective reports indicated a predicted enhancement in liking across increasing interest levels. Electroencephalography (EEG) recordings during image viewing revealed different patterns of alpha power in temporal brain regions across interest levels. Experiment 2 enforced a brief initial image-viewing stage and a subsequent reflective image-viewing stage. Differences in alpha power arose in most EEG channels between the initial and deliberative viewing stages. A linear increase in aesthetic liking was again seen across interest levels, with different patterns of alpha activity in temporal and occipital regions across these levels. Overall, the phenomenological data support the PIA Model, while the physiological data suggest that enhanced aesthetic liking might be associated with “flow-feelings” indexed by alpha activity in brain regions linked to visual attention and reducing distraction.

## 1. Introduction

The experimental investigation of aesthetic appreciation has a long history in psychological research, deriving from the foundational work of [40] ([40]), who drew links between the objective properties of stimuli (e.g., their perceptual symmetry) and people’s judgments of beauty. This theme was subsequently taken forward by [15] (e.g., [15], [16]), whose research included a focus on the physiological mechanisms that mediate between objective stimulus properties and aesthetic responses, including people’s arousal states. In more recent years, Berlyne’s conceptual advancements in the study of aesthetic appreciation have informed contemporary theorizing, including “processing-fluency” approaches to explaining aesthetic pleasure (e.g., see [6]; [82]). Such approaches propose that people’s aesthetic judgments are rooted in the processing dynamics associated with *perception*. More specifically, it has been argued that the subjective ease with which mental operations are performed when perceiving an image or object gives rise to processing experiences that are “hedonically marked” ([82]; [102]), such that ease of processing equates to a pleasurable experience. In other words, this “Hedonic-Marking Theory” captures the principle that people prefer easily processed stimuli because the affective response is positively valenced.

Evidence for a direct, causal connection linking perceptual fluency to aesthetic liking derives from numerous experimental studies involving phenomenological self-report measures, as well as from studies examining psychophysiological indicators of positive affect. For example, research involving electromyography to measure the facial muscle activity that arises during aesthetic judgments that are driven by perceptual fluency has demonstrated stronger activity in the facial zygomaticus region, which is associated with smiling, but not in the corrugator region, which is associated with frowning ([94]; [100]). It is also noteworthy that perceptual-fluency theories of aesthetic liking generally assume that *multiple* stimulus factors can feed into a global experience of processing ease or difficulty (e.g., [6]; [21]; [52]; [64]). Indeed, a range of stimulus sources have been linked to highly fluent perceptual experiences, including visual symmetry ([18]; [49]), visual clarity ([80]), figure–ground contrast ([83]), curvature of contours ([9]), exposure duration ([83]) and perceptual priming ([83]). In addition, prototype formation ([101]) and repeated exposure ([104]) have also been found to make a stimulus easier to process.

Although the Hedonic-Marking Theory of aesthetic liking dominated early theorizing in this area, it is noteworthy that other fluency accounts have also been influential for conceptual advancement, including ones proposing that people’s fluency interpretations are based on the application of “naïve theories” regarding the interpretation of fluency and disfluency cues. Such “Cue-Attribution” accounts entail a level of deliberative or reflective thinking and causal attribution, which means that people’s judgments are potentially malleable (see [6]; [88]). In other words, although the experience of processing fluency serves as a general “metacognitive cue” during thinking, reasoning and decision making, the *interpretation* of experienced fluency is flexible and may differ from context to context. For example, instructional changes ([22]) indicate that it is possible to manipulate the naïve theories that people apply when making judgments, thereby reversing the default interpretation of high fluency being associated with pleasure and aesthetic liking.

### 1.1. Stimulus Complexity and Aesthetic Preference

A key debate in perceptual fluency research concerns the nature of the relationship between stimulus *complexity* and aesthetic preference, where stimulus complexity is typically operationalized in terms of the number and variety of elements that are present in a visual scene or an image (e.g., [14]; [55]; [72]; [95]). Hedonic-Marking Theory ([82]; [102]) would predict the existence of a linear relationship between complexity and liking, with increasingly complex stimuli giving rise to decreased liking. Although this relationship has occasionally been observed (e.g., [37]), stimulus simplicity does not always lead to more positive affect. Indeed, some studies have instead shown a positive correlation between complexity and liking (e.g., [71]), whereas other studies have uncovered an inverted U-shaped relationship between these variables, with intermediate-complexity stimuli showing higher preference judgments compared to either low-complexity or high-complexity ones (e.g., [15]; [77]; [98]). There is likewise much evidence that people can derive considerable aesthetic pleasure from viewing complex images or visually challenging objects or scenes (e.g., [7]; [55]; [66]; [73]), including ambiguous pictures ([53]), fractals ([55]) and complex visual artworks ([58]). Moreover, in an analysis of variables that increased aesthetic pleasure, [73] ([73]) found that more complex stimuli were liked more than simpler stimuli because they were considered to be more *meaningful*, pointing to a potential role for “conceptual fluency” in aesthetic liking judgments (cf. [93]; [99]), which is a theme we return to later.

Such a mixed array of findings regarding the relationship between stimulus complexity and aesthetic preference presents something of a challenge for processing-fluency theories of aesthetic liking, and several explanations have been advanced in an effort to explain the effects that have been observed. For example, [82] ([82]) have argued that low levels of complexity might make the “source” of the fluency experience highly salient, thereby suppressing the normal preference for high-fluency stimuli. As complexity increases, however, [82] ([82]) propose that the salience of the source of fluency will decrease, thereby enhancing aesthetic liking. Further increases in complexity will eventually impair processing fluency and thus will reduce aesthetic liking. In this way, an inverted U-shaped relation between stimulus complexity and aesthetic liking might arise in the case of complexity manipulations that make the source of experienced fluency highly salient at the simpler end of the complexity continuum.

Much harder to explain from a processing-fluency perspective, however, is the evidence for people deriving increasing aesthetic pleasure when viewing highly complex and sophisticated images, objects, scenes and artworks (e.g., [7]; [55]; [58]; [66]; [73]). Such conceptual challenges for traditional processing-fluency accounts have led to the development of new theories of fluency-based aesthetic liking that represent a significant departure from more established views. One such account is the “Pleasure-Interest Model of Aesthetic Liking” (PIA Model) proposed by [42] ([42]), which we overview in the next section in order to demonstrate how it can explain seemingly paradoxical findings regarding the relationship between increasing complexity and elevated aesthetic liking, while also lending itself to the derivation of novel, testable predictions.

### 1.2. The Pleasure-Interest Model of Aesthetic Liking (PIA Model)

[42] ([42]) PIA Model offers a “dual-process” perspective on issues relating to fluency-based aesthetics and represents a direct attempt to reconcile inconsistent findings in the literature relating to aesthetic preference judgments. According to the PIA Model, aesthetic preferences arise from two distinct fluency-based processes that take place sequentially: (1) an immediate, automatic and default *percept-driven* process that occurs upon encountering an aesthetic object, which gives rise to an initial aesthetic judgment of pleasure or displeasure; and (2) a subsequent *perceiver-driven*, deliberative or reflective process, which is initiated because of a person’s motivation to process a stimulus further, giving rise to a fluency-based aesthetic evaluation of interest, boredom or confusion. An especially compelling aspect of [42] ([42]) dual-process theory is, therefore, its capacity to capture the interplay between initial percept-driven judgments and perceiver-driven override of such default judgments.

In developing their PIA Model, [42] ([42]) draw extensively upon the work of other theorists (e.g., [7]; [23]; [78]; [79]), who propose that a perceiver’s active cognitive *elaboration* of a stimulus can play a key role in aesthetic liking. According to such views, a perceiver does not merely react passively to a stimulus; they can also engage actively with it, devoting additional processing effort toward the stimulus to gain a deeper interpretation and understanding of it. Experimental evidence supports the importance of active cognitive elaboration in influencing aesthetic liking. For example, studies using paintings as stimuli and experimental manipulations relating to the presence or absence of titles or accompanying descriptive or stylistic information have revealed that aesthetic appreciation can be enhanced through elaborative processing, so long as such processing is associated with a *meaningful* analysis of the presented stimuli (e.g., [11]; [68]; [75]; [87]). Furthermore, elaborative processing that is triggered by instructing participants to evaluate stimuli on multiple dimensions has been shown to engender an enhanced appreciation of product designs such as car exteriors, but only when these designs are novel, innovative or atypical (see [24]; [23]; [38]; [67]).

[42] ([42]) argue that the aforementioned findings give good grounds for the existence of a positive relationship between elaborative processing and aesthetic liking in situations where the stimulus holds what they refer to as an “appropriate elaboration affordance”, which will often arise in situations where initial *disfluent* processing has occurred. Graf and Landwehr further argue that inconsistent findings relating to the association between processing fluency and liking are attributable to the failure of the standard processing-fluency theory to recognize that perceivers can take an active, reflective role in processing a stimulus that is initially processed disfluently in order to override the early experience of disliking and replace it with a judgment of liking. The PIA Model is, therefore, able to capture the idea that whilst aesthetic liking can arise from an initial default process based on stimulus-driven cognitive operations that occur automatically and mandatorily, it is nevertheless possible that aesthetic liking arising from perceiver-driven deliberative elaboration will have an opposite valence to the default experience. [42] ([42]) conceptualize this second, reflective process as involving “higher order” cognitive operations, such as careful analysis and interpretation of a stimulus, including the assignment of meaning to it.

We have previously noted (see [8]) the close alignment between [42] ([42]) dual-process theory of aesthetic liking and research on *meta-reasoning* in the literature on judgment, decision making and reasoning (e.g., [3], [4]; see also [84]; [85]). Meta-reasoning research has also traditionally adopted a dual-process stance, emphasizing how metacognitive monitoring processes are sensitive to a variety of cues (see [2]), including ones that derive from perceivable features of the task (e.g., its apparent complexity), as well as ones that derive from the experience of attempting the task (e.g., ease of processing). [3] ([3], [4]) “meta-reasoning framework”—much like Graf and Landwehr’s PIA Model—also captures the idea that initial disfluent processing in problem-solving and decision-making contexts may trigger more reflective, analytic processing. As such, [42] ([42]) PIA Model complements developments in the field of meta-reasoning, further supporting the model’s credibility. In the next section, we consider some additional, core assumptions of the PIA Model and evidence to support them. Doing this will take us a step closer to the aim of the research that we report in the present paper, which is concerned with examining the link between the “interestingness” of artistic stimuli and people’s phenomenological experience of pleasure, as well as the psychophysiological correlates of this experience.

### 1.3. The PIA Model and the Interplay between Stimulus Complexity and Conceptual Fluency

Many important conceptual ideas are encapsulated within [42] ([42]) theorizing (see also [43]), which give the PIA Model its considerable predictive power. Here, we examine three underpinning assumptions of the model that were of particular relevance in motivating the predictions that we set out to test experimentally in the present research. The first assumption to note relates to the concept of “conceptual fluency”, as mentioned earlier, which refers to the ease of deriving *meaning* from a stimulus ([93]; [99]). [42] ([42]) explain that both initial default processing and subsequent reflective processing can be influenced by conceptual fluency as well as by perceptual fluency. Importantly, however, they claim that default processing will be relatively more influenced by perceptual fluency and that deliberative processing will be relatively more influenced by conceptual fluency. The rationale behind this claim is that perceptual fluency is a passive, automatic, stimulus-driven experience, whereas conceptual fluency is an active, reflective, perceiver-driven process that places a substantial burden on elaborative and interpretative reasoning. This assumption implies that stimulus *complexity* (i.e., a stimulus-based property) will primarily have an impact at the default processing stage, with higher complexity promoting an increased sense of disfluency. The assumption additionally implies that *conceptual fluency* (arising from the ease of meaning extraction and stimulus interpretation) will primarily have an impact at the reflective processing stage.

The second noteworthy assumption of the PIA Model is that the cue to move from default processing to reflective processing is considered to be jointly determined by an interplay between two factors: first, a feeling of disfluency, which signals to the perceiver the need to invest more effort in processing the stimulus; and second, the perceiver’s need for “cognitive enrichment”. What this means is that if a person experiences disfluency during the default processing stage and has a high need for cognitive enrichment, then the motivation to engage in reflective processing will be especially strong. In situations where disfluency and the need for cognitive enrichment are in opposition to one another, then it is the relative strength of these factors that will determine whether reflective processing is triggered.

The third assumption of the PIA Model that has relevance to the present research concerns the manner in which reflective processing can give rise to aesthetic evaluations. As we have mentioned previously, what is initially a disfluently processed stimulus may subsequently be found to be relatively easy to integrate into existing knowledge structures when processed reflectively (i.e., it is conceptually fluent). The updated fluency level that is experienced after reflective processing can thereby lead to a final aesthetic evaluation that is far more positive than the initial aesthetic evaluation that arose from default processing.

In combination, these aforementioned assumptions lead to some intriguing predictions, which were tested experimentally in a study reported by [8] ([8]) that simultaneously manipulated the complexity (low vs. high) of presented stimuli (i.e., abstract artworks) and their conceptual fluency (across five linearly increasing levels ranging from low to high). In their study, [8] ([8]) first predicted that there should be a main effect of conceptual fluency on aesthetic liking such that conceptually fluent stimuli should be liked to a greater extent than conceptually disfluent stimuli. This prediction reflects the assumption that being more readily able to derive a meaningful interpretation from a presented stimulus should be a relatively pleasurable experience. Second, it was predicted that the effect of conceptual fluency on aesthetic liking should serve to modulate the impact of stimulus complexity, giving rise to a complexity by conceptual fluency interaction. This predicted interaction reflects the PIA Model’s assumption that it is complex stimuli (i.e., those that are relatively disfluent at the default processing level) rather than simple stimuli (i.e., those that are relatively fluent at the default processing level) that should trigger more reflective processing and effort after meaning. The consequence of complex stimuli being subjected to such reflective processing is that they should be associated with increased aesthetic liking compared to simpler stimuli, so long as they are also conceptually fluent. In other words, people should tend to have enhanced liking for abstract artworks that initially seem to be complex, but which then turn out to be relatively easy to derive meaning from. In contrast, abstract artworks that initially seem to be complex and that then remain hard to derive meaning from should persist in being fairly unappealing.

[8] ([8]) found good evidence to support their predictions. First, a significant main effect of conceptual fluency was observed on beauty ratings, with the data indicating the presence of a highly reliable linear trend whereby abstract artworks were judged to be progressively more beautiful at increasing levels of conceptual fluency. Second, the analyses revealed the existence of a significant main effect of complexity on beauty ratings, with high-complexity artworks being rated as more beautiful than low-complexity artworks. Third, and most importantly, the predicted interaction was found to be significant, with evidence indicating that stimulus complexity modulated the effect of conceptual fluency. Follow-up tests revealed significant differences in beauty ratings between high- versus low-complexity artworks across the three highest levels of conceptual fluency but no significant differences in beauty ratings between high- versus low-complexity artworks at the two lowest levels of conceptual fluency. In sum, [8] ([8]) findings fully support the assumptions of the PIA Model by demonstrating that people like more complex abstract artworks compared to simpler ones, but only when they can readily derive meaning from such apparently complex stimuli. In cases where the extraction of meaning is more elusive, people show reduced liking for abstract artworks and no separation in liking between complex versus simpler pieces.

As part of the present research, we also wished to go beyond the core assumptions of the PIA Model so as to explore the potential links between people’s experience of engaging in aesthetic appraisals involving enhanced liking and the phenomenological experience of being in in a *flow-like* state (see [30]; [31]). The concept of *flow* refers to a positively valenced affective mental state that is characterized by complete concentration and absorption in a specific task in the present moment ([27]; [92]). More specifically, [29] ([29]) and [30] ([30]) identified eight main characteristics of flow states: (1) challenge and skill balance; (2) clear goals; (3) automaticity and immediate feedback; (4) intense concentration; (5) time distortion; (6) the paradox of control; (7) loss of self-consciousness; and (8) self-rewarding autotelic experiences. Interestingly, [28] ([28]) also introduced the concept of “microflow”, which he believed could arise during activities such as the observation of artworks that require a relatively low level of skill and challenge and that are less intensive and complex (see also [29]; [30]). We contend, however, that these proposals miss the critical role that conceptual fluency can play in modulating the effects of complexity, as predicted by the PIA Model and as demonstrated empirically by [8] ([8]). In the present study, we therefore assumed that presented artworks that give rise to more positive aesthetic appraisals through a combination of complexity and conceptual fluency might also give rise to higher subjective ratings of flow-like experiences.

### 1.4. Aims of the Research

The present research aimed to provide a conceptual replication of key aspects of [8] ([8]) study to provide further phenomenological evidence in support of important assumptions associated with [42] ([42]) PIA Model relating to the basis of aesthetic liking. The research also afforded an opportunity to examine whether the aesthetic experience of interest-based pleasure, as predicted by the PIA Model, is associated with the subjective experience of being in a flow-like state. In addition, the research aimed to determine whether there is a specific psychophysiological signature that is detectable in brain activity (measured using electroencephalography (EEG)), arising when a person is experiencing interest-based pleasure and entering a flow-like state.

The research that we report took as its starting point one of the key ideas encapsulated within [42] ([42]) PIA Model, which is that certain stimuli have characteristics that give rise to enhanced levels of “interest” for the perceiver, thereby promoting both increased reflective processing as well as the potential for increased aesthetic liking. As we have noted, such theorizing is supported by [8] ([8]) empirical results, where it was found that for presented images the combination of heightened visual complexity (a perceptual property) and heightened conceptual fluency (a perceiver-driven process) appears to promote increased interest, which manifests behaviorally as elevated aesthetic liking (presumably because high conceptual fluency is hedonically marked as positive).

In the present study, we therefore directly set out to present participants with visual stimuli at three levels of interest: low interest (i.e., low complexity and low conceptual fluency); moderate interest (i.e., moderate complexity and moderate conceptual fluency); and high interest (high complexity and high conceptual fluency). We describe in detail in the next section the process that we pursued to categorize images across these three levels of interest. Rather than using abstract art, as in [8] ([8]) study, we decided instead to use graffiti street art, both to ensure participants’ likely lack of familiarity with presented stimuli and to generalize key aspects of Ball et al.’s previous findings to a different yet highly contemporary artistic medium. Our primary behavioral prediction was that people’s subjective ratings of liking in relation to presented artworks should increase linearly across these three levels of interest (cf. [8]). Our secondary behavioral prediction related to our assumption that more positive aesthetic appraisals should also give rise to higher subjective ratings of flow-like experiences.

Our final prediction concerned the neurological underpinnings of positive aesthetic appraisal and associated flow-like experiences. At a physiological level, the experience of flow has been linked to increased alpha brain waves (8–12 Hz), which are associated with being in a relaxed yet alert mental state (e.g., [65]). A peak in relation to alpha activity, referred to as an “alpha peak”, has been found during object recognition and visual encoding ([61]) and sustained visual attention (e.g., [5]; [62]), as well as during the application of executive functions ([81]), including working memory ([70]). Moreover, increased alpha power has also been linked to aesthetic judgments of beauty, as well as positive emotional states in the brain’s frontal region ([25]), and has additionally been found to be implicated in aesthetic appraisals of artworks ([26]). In sum, a body of research suggests that alpha brain rhythms are linked both to flow experiences and to aesthetic appraisals, suggesting that increased alpha activity represents a psychophysiological marker in research on pleasure responses to presented stimuli. As such, our final prediction related specifically to the EEG correlates of more positive aesthetic appraisals and our expectation that such appraisals would be associated with increased alpha activity across cortical regions relative to alpha activity arising in relation to less positive aesthetic appraisals.

## 2. Experiment 1

The aim of Experiment 1 was to test our phenomenological and psychophysiological predictions by presenting participants with images of graffiti street art that varied systematically across three levels of interest. As noted above, to establish images that should be experienced as having low interest, we needed to identify street-art stimuli that were of low complexity and low conceptual fluency, which are henceforth referred to as LCLF stimuli. Likewise, to establish images of moderate interest, we needed to identify street-art stimuli that were of moderate complexity and moderate conceptual fluency (henceforth, MCMF stimuli), and to establish images of high interest, we needed to identify street-art stimuli that were of high complexity and high conceptual fluency (henceforth, HCHF stimuli). To pre-categorize images according to these criteria, we conducted a preliminary study with 195 photographic images of street art (see the Appendix A), which were presented to participants sequentially, with self-report ratings being acquired for each image of either its complexity or its conceptual fluency, depending on the instructions given to participants. Additionally, the average and maximum appraisal times were measured for each image to inform our subsequent experiments.

To implement this image pre-categorization study, we used the Prolific Academic online data-collection platform, with the participant pool being recruited from various parts of the world, including Brazil, Canada, Germany, Greece, Italy, Jamaica, Mexico, Poland, Portugal, the United Kingdom and the United States. All participants were fluent in English and either had no visual impairments or had corrected-to-normal vision. Individuals were excluded from the study if they were under the age of 16, if they were vulnerable adults with learning disabilities or if they were adults with mild cognitive impairments. We recruited separate samples of participants for the complexity ratings and for the conceptual fluency ratings to avoid any possibility that participants’ ratings might in some way become cross-dependent. Participants were rewarded at the standard rate for taking part in the study, which equated to the UK minimal wage at the time of data collection.

To acquire complexity ratings for presented images, we recruited 101 participants (age range: 18–45 years, *M* = 27.29, *SD* = 7.90; 53 males, 48 females). Participants registered their complexity ratings for all 195 images using Qualtrics XM software (https://www.qualtrics.com/uk/), which drove stimulus presentation and response collection. Participants were asked to appraise each image within 30 s in response to the question, “How complex was the image to you?”, and to mark their complexity rating on a scale that ranged from 0 (*not at all complex*) to 100 (*very complex*), in accordance with the method used by [8] ([8]).

To obtain conceptual fluency ratings for presented images, we recruited 99 participants (age range: 18–45 years, *M* = 28.05, *SD* = 7.43; 48 male, 50 female, one undisclosed gender). Again, these participants used Qualtrics XM software to register their conceptual fluency ratings for the 195 images. Participants were required to appraise each image within 30 s in response to the question, “How meaningful was the image to you?”, and to mark their conceptual fluency rating on a scale that ranged from 0 (*not at all meaningful*) to 100 (*very meaningful*), in accordance with the method deployed by [8] ([8]).

The first step in progressing toward a categorization of the 195 presented images based on their complexity and conceptual fluency ratings involved creating Z-scores for each rating to standardize participants’ judgments. We next conducted exploratory curve-fit analyses using the Z-scores to check for a linear relationship between complexity ratings and conceptual fluency ratings for all 195 images (see the Appendix A for further details). It was found that the linear curve fit, *F*(1, 193) = 138.695, *MSE* = 81.119, *p* < .001, was stronger than both the quadratic curve fit, *F*(2, 192) = 70.575, *MSE* = 41.097, *p* < .001, and the cubic curve fit, *F*(3, 191) = 47.007, *MSE* = 27.466, *p* < .001. The high degree of linear association between the complexity and the conceptual fluency ratings justified undertaking a correlation analysis of the Z-transformed complexity and conceptual fluency ratings, which revealed that complexity and conceptual fluency were highly positively correlated, *r* = 0.647, *p* < .001 (two-tailed), supporting the viability of combining these ratings to form a composite categorization of a sub-set of the 195 images in terms of three specific levels of interest: LCLF, MCMF and HCHF. For clarity, we also note here that the highly correlated nature of the complexity and conceptual fluency ratings meant that it would *not* have been possible to select images to allow for the formulation of a full factorial design, similar to that reported by [8] ([8]), involving the simultaneous manipulation of image complexity across multiple levels (e.g., low, moderate and high) and conceptual fluency across multiple levels (e.g., low, moderate and high).

To create a sub-set of images across three levels of interest for use in our experiment, we applied three image-categorization rules. First, images with standardized scores of −1.00 or lower for both complexity and conceptual fluency were categorized as “low complexity, low conceptual fluency” (LCLF). Second, images with standardized scores of between −0.20 and +0.20 for both complexity and conceptual fluency were categorized as “moderate complexity, moderate conceptual fluency” (MCMF). Third, images with standardized scores of +1.00 or higher for both complexity and conceptual fluency were categorized as “high complexity, high conceptual fluency” (HCHF). Images outside of these ranges were not considered further. After the categorization stage, nine images were selected randomly from their respective category (i.e., LCLF, MCMF and HCHF) to form three image sub-sets that were suitable for use in Experiment 1. We present an example image from each category in Figure 1. The overall mean viewing time for each image across all conditions was 7.96 s (*SD* = 0.53). More specifically, participants spent a mean of 7.69 s (*SD* = 0.57) gazing at the selected LCLF images, 8.08 s (*SD* = 0.58) gazing at the selected MCMF images and 8.12 s (*SD* = 0.36) gazing at the selected HCHF images.

### 2.1. Method

#### 2.1.1. Design

The experiment involved a repeated-measures design with one independent variable, level of interest, captured by the three image categories: LCLF, MCMF and HCHF. The dependent variables were the subjective ratings of aesthetic liking, complexity, conceptual fluency, overall perception of flow, concentration, time distortion, arousal and pleasantness, as well as EEG measures of alpha power across brain regions.

#### 2.1.2. Participants

The experiment took place in a research laboratory and recruited participants who had not been involved in the image pre-categorization study. The sample size was calculated through an a priori power analysis (*d* = 0.60, 1 − β = 0.80, α = 0.05) using G*Power 3.1.9.4 ([39]), with the expected effect size informed by previous research on the neural markers of peak-performance experiences ([19]). Sixteen participants were recruited (eight males, eight females; age range: 18–45 years, *M* = 23.69, *SD* = 4.74). The same inclusion and exclusion criteria were applied as in the pre-categorization study. Participants were rewarded at the standard rate for taking part in the study, which equated to the UK minimal wage at the time of data collection.

#### 2.1.3. Materials

The experimental task involved presenting participants with a sequence of 27 images of graffiti street art via PowerPoint and requesting them to provide subjective ratings on various dimensions for each image, while EEG recordings were also taken. The nine images at each level of interest (categorized as LCLF, MCMF and HCHF) were presented to participants in a block. The order of the three blocks of images was counterbalanced and the presentation order of images within each block was randomized. At the end of the first and second blocks, a 5 min break was provided. Before and after the presentation of each image, a blank white screen was displayed in the inter-trial interval for 3 s (cf. [96]; [103]).

#### 2.1.4. Procedure

Participants were provided with a briefing regarding the procedure for the study and signed an informed consent form. The EEG cap was then applied. A baseline EEG measure was taken, which lasted for 4 min (2 min with eyes closed and 2 min with eyes open). Taking this baseline EEG measure also enabled a check to be made that the EEG equipment was working correctly and that the EEG oscillations were within the expected range. Each participant was then presented with the sequence of 27 street-art images on a 44.17 × 23.77 inch monitor screen, with image presentation time locked in accordance with the EEG recording markers using strict timings for the PowerPoint presentation. Each image involved an imposed viewing time of 8 s, in line with the viewing time norms established in the pre-categorization study. This time window also coincides with previous research suggesting that a 6–15 s stimulus presentation window can be considered optimal for the analysis of bio-signal data (e.g., [60]; [103]).

Following the presentation of each image, participants were asked to report their subjective rating of the image in terms of aesthetic liking, complexity and conceptual fluency. Aesthetic liking ratings were obtained by asking participants to answer the question, “How much did you like the image?”, with responses being registered on a scale ranging from 0 (*not at all*) to 10 (*very much*), in line with the procedure adopted by [44] ([44]). Scores for complexity and conceptual fluency were elicited using a 100-point scale, as in the pre-categorization study.

Flow experiences following the presentation of each image were elicited using items from the “Short Flow State Scale” and the “Core Dispositional Flow Scale” ([50], [51]) to measure overall perceptions of flow, concentration and time distortion. Regarding the overall perception of flow (Item 4 from the Core Dispositional Flow Scale), participants were asked to respond to the question, “Did you feel like you were ‘in the zone’ while you were gazing at the image?” using a scale that ranged from 0 (*not at all*) to 10 (*very much*). With respect to concentration (Item 5 from the Short Flow State Scale), participants were asked to respond to the question, “How focused did you feel while looking at the image?” using a scale that ranged from 0 (*not at all*) to 10 (*very much*). In relation to time distortion (Item 8 from the Short Flow State Scale), participants were asked to respond to the question, “Did you feel time passing at a different pace while you were looking at the image?” using a scale ranging from 0 (*not at all*) to 10 (*very much*).

In using only three items from these established flow scales, we acknowledge that concerns might be raised regarding the way in which our selective approach to measuring flow in the present study may have threatened the construct validity and reliability of the original measurement instruments. We accept that this is a fair criticism, although we also believe that it would have been inappropriate to deploy the full set of items from these established scales to assess flow experiences for *each* of the 27 images presented in our study, not least because these scales were not designed to examine perceived flow during the brief presentation of a multiplicity of changing visual stimuli. We also note that other items from these scales, such as those relating to the possession of clear goals, to challenge and skill balance, to automaticity and immediate feedback and to the paradox of control, were much less relevant to the context of aesthetic appraisal that pertained to the present study. We additionally emphasize that single-item measures are often viewed as a reliable way to measure cognitive–affective states in applied psychology (e.g., [89]).

In the present study, arousal and pleasantness states were also measured following the presentation of each image. This was done using an adapted version of an affect grid ([34]; [86]). Participants were asked to respond to the question, “How activated did you feel while looking at the image?” using a scale ranging from 0 (*total sleepiness*) to 10 (*highly activated*). Additionally, participants were asked to respond to the question, “How pleasant/enjoyable was it to look at the image?” using a scale ranging from 0 (*not at all pleasant*) to 10 (*highly pleasant*).

Throughout the experimental procedure, EEG brain waves were recorded using a NeXus-32 biofeedback system ([76]). Specifically, alpha (8–12 Hz) absolute power was measured in microvolts squared (µV^2^) across 21 electrodes at a sampling frequency of 256 Hz. The electrodes were positioned over the scalp and followed the 10/20 system ([1]). The ground electrode was located at channel Afz, between channels Fpz and Fz. Impedance values of Z < 10 kΩ were maintained during data collection.

Once the study was completed, participants were debriefed, thanked for their time and given a chance to ask questions. The full duration of the study was approximately two hours.

### 2.2. Results

First, the results relating to the phenomenological data will be presented before we present the psychophysiological results from the EEG analysis. For the phenomenological data, means, standard deviations, *F*-values, *p*-values and effect-size measures are reported in Table 1, in accordance with current reporting standards. All subjective data were analyzed using repeated-measures analysis of variance (ANOVA), with Bonferroni adjustments applied for all post hoc comparisons.

#### 2.2.1. Aesthetic Liking, Conceptual Fluency and Complexity

A repeated-measures ANOVA indicated that participants’ subjective ratings of aesthetic liking were significantly different across levels of interest, *F*(1, 286) = 40.50, *p* < .001; *η_p_*^2^ = 0.22, with follow-up tests showing higher ratings for HCHF images than for either MCMF images (*d* = 0.44) or LCLF images (*d* = 1.04), as well as higher ratings for MCMF images than LCLF images (*d* = 0.36). These results corroborate the view that more complex and conceptually fluent images are liked more than images of lower complexity and conceptual fluency. Repeated-measures ANOVAs were also conducted on the complexity and conceptual fluency ratings by way of a manipulation check. The analysis of the complexity data revealed that participants’ subjective ratings were significantly different across levels of interest, *F*(1, 286) = 54.21, *p* < .001; *η_p_*^2^ = 0.28, with complexity ratings showing a predicted pattern of differences across interest levels in follow-up tests; that is, higher ratings for HCHF images than for either MCMF images (*d* = 0.53) or LCLF images (*d* = 1.24), as well as higher ratings for MCMF images than LCLF images (*d* = 0.71). Similarly, the analysis of the conceptual fluency data revealed that participants’ subjective ratings were significantly different across levels of interest, *F*(1, 286) = 50.09, *p* < .001; *η_p_*^2^ = 0.26, with conceptual fluency ratings showing a predicted pattern of differences across interest levels in follow-up tests; that is, higher ratings for HCHF images than for either MCMF images (*d* = 0.50) or LCLF images (*d* = 1.01), as well as higher ratings for MCMF images than LCLF images (*d* = 0.54).

#### 2.2.2. Overall Perception of Flow, Concentration and Time Distortion

A repeated-measures ANOVA indicated that participants’ overall perception of flow was significantly different across levels of interest, *F*(1, 286) = 36.80, *p* < .001; *η_p_*^2^ = 0.21, with follow-up tests showing higher ratings for HCHF images than for either MCMF images (*d* = 0.81) or LCLF images (*d* = 0.81), as well as higher ratings for MCMF images than LCLF images (*d* = 0.52). Other measures of flow-like experiences showed broadly equivalent effects, albeit with reduced effect sizes. The analysis of participants’ self-rated concentration was significantly different across levels of interest, *F*(1, 286) = 11.33, *p* < .001; *η_p_*^2^ = 0.07, with follow-up tests showing higher ratings for HCHF images than for either MCMF images (*d* = 0.31) or LCLF images (*d* = 0.41), as well as higher ratings for MCMF images than LCLF images (*d* = 0.13). Likewise, the analysis of participants’ self-rated time distortion was significantly different across levels of interest, *F*(1, 286) = 9.30, *p* < .001; *η_p_*^2^ = 0.06, with follow-up tests showing higher ratings for HCHF images than for either MCMF images (*d* = 0.27) or LCLF images (*d* = 0.45), as well as higher ratings for MCMF images than LCLF images (*d* = 0.16).

#### 2.2.3. Arousal and Pleasantness

The final analyses of participants’ phenomenological experiences focused on their ratings relating to their subjective states of arousal and pleasantness having viewed a presented image. These data were found to align with the established pattern of evidence associated with other subjective ratings that we have reported. With respect to feelings of arousal, a repeated measures ANOVA showed that participants’ sense of activation was significantly different across levels of interest, *F*(1, 286) = 15.81, *p* < .001; *η_p_*^2^ = 0.10, with follow-up tests indicating higher ratings for HCHF images than for either MCMF images (*d* = 0.32) or LCLF images (*d* = 0.47), as well as higher ratings for MCMF images than LCLF images (*d* = 0.16). For reported feelings of pleasantness, a repeated-measures ANOVA showed that participants’ feelings of pleasantness were significantly different across levels of interest, *F*(1, 286) = 25.92, *p* < .001; *η_p_*^2^ = 0.15, with follow-up tests indicating higher ratings for HCHF images than for either MCMF images (*d* = 0.44) or LCLF images (*d* = 0.77), as well as higher ratings for MCMF images than LCLF images (*d* = 0.35).

#### 2.2.4. EEG Findings Relating to Alpha Power

All EEG data were visually inspected, filtered and exported using the functions built into the BioTrace+ software (https://www.mindmedia.com). The data were segmented into the 8 s time windows during which each image had been presented. Univariate outlier analysis was carried out in line with current multivariate statistical guidelines ([45]), with absolute Z-score values above 2.5 being removed from the dataset. The data were separated according to each frequency band, alpha (8–12 Hz), beta (16–24 Hz) and theta (4–8 Hz), although only data for alpha were analyzed further for the three levels of interest relating to the categorized images (LCLF, MCMF and HCHF). EEG data were analyzed using repeated-measures ANOVAs, with Bonferroni adjustments applied for all post hoc comparisons. We note in advance that although alpha power was sampled at 19 key electrode sites, we only report in this section the findings from three regions (T3, T5 and T6), as these were the only regions to reveal significant differences across the image categories.

Topographical heat maps were created relating to the EEG electrode sites based on the absolute alpha power values and are presented in Figure 2A. Examination of these heat maps reveals a variety of alpha-wave activity patterns across the three interest levels for categorized images: LCLF, MCMF and HCHF. Figure 2B presents the *p*-values and Cohen’s *d* effect-size values for the electrode sites that revealed significant differences across image categories. Alpha power showed a significant difference across image categories at T3, *F*(1.88, 136.92) = 12.28, *p* < .001, *η_p_*^2^ = 0.14. Post hoc comparisons revealed that alpha power was highest during the viewing of HCHF images relative to MCMF images (*p* < .001, *d* = 0.61). Alpha power also showed a significant difference across levels of interest at T5, *F*(1.88, 136.92) = 6.75, *p* < .005, *η_p_*^2^ = 0.09. Post hoc comparisons revealed that alpha power was highest during the viewing of LCLF images relative to HCHF images (*p* < .005, *d* = −0.51). Finally, alpha power showed a significant difference across levels of interest at T6, *F*(1.88, 136.92) = 6.44, *p* < .005, *η_p_*^2^ = 0.07. Post hoc comparisons revealed that alpha power was highest during the viewing of HCHF images relative to LCLF images (*p* < .005, *d* = 0.42).

### 2.3. Discussion

Previous research relating to aesthetic preferences that has been informed by the PIA Model (e.g., [42]) has demonstrated that complex images that are also conceptually fluent (i.e., meaningful) can lead to increased liking judgments relative to simpler images, irrespective of the conceptual fluency of the latter (see [8]). In line with the PIA Model, we suggest that complex yet conceptually fluent images (e.g., artworks) can promote higher levels of interest in the perceiver than images that are less complex and less conceptually fluent. The present experiment manipulated the presentation of images of graffiti street art across three levels of interest: low complexity and low conceptual fluency (LCLF); moderate complexity and moderate conceptual fluency (MCMF); and high complexity and high conceptual fluency (HCHF). In line with predictions, HCHF images were found to promote increased phenomenological ratings of aesthetic liking relative to MCMF images, which in turn were liked more than LCLF images. Importantly too, the same pattern of subjective ratings across the image categories was seen for experiences of flow, concentration and time distortion, as well as for experiences of arousal and pleasure. These latter findings establish a potentially important link between theoretical constructs such as complexity and conceptual fluency that are associated with the experience of aesthetic liking and the concept of flow, as discussed by [29] ([29]; see also [30]).

Of further importance are our psychophysiological findings arising from our EEG analyses with respect to differences in alpha power when participants were viewing images at different interest levels. Increased alpha power has been found to be related to sustained visual attention ([5]; [62]). It is, therefore, of interest that the T3 electrode site, which is broadly related to the ventral attention network, visual perception and memory-encoding processes ([13]; [59]), revealed increased alpha power when participants were viewing HCHF images relative to MCMF images. This finding suggests that participants were finding HCHF images more visually interesting relative to MCMF images, which is in line with what would be expected according to the PIA model. Regarding differences that were seen in alpha power across image categories at electrode site T6, we note that the right temporal region is important in visual memory, in interpreting the meaning of body language, in understanding social cues and in object recognition ([46]; see also [17]). Increased alpha power at T6 during the viewing of HCHF artworks relative to those in the LCLF category may, therefore, suggest that participants were recruiting resources necessary for visual recollection so as to make meaningful interpretations of HCHF images.

The present experiment additionally found increased alpha power at the T5 electrode site when participants were viewing images in the LCLF category relative to the HCHF category. The left temporal lobe, particularly Wernicke’s area in T5, is involved primarily with speech and language comprehension ([10]; [97]). This region involves transferring visual stimuli into semantic categories from language (e.g., [32]; [48]; [47]). When viewing images in the LCLF category, participants may have been using more resources to attribute semantic meaning to the images (i.e., to “put them into words”) because they were less meaningful than images in the HCHF category, where alpha power was significantly lower.

## 3. Experiment 2

The aim of Experiment 2 was to test more directly the dual-process assumptions that underpin [42] ([42]) PIA Model, which proposes that people engage in two processing stages: an initial automatic, *percept-driven* default process and a subsequent *perceiver-driven* reflective process. Furthermore, stimulus complexity is viewed as being a key trigger for people engaging in the second, reflective processing stage, as perceivers are likely to be motivated to apply elaborative reasoning to explore complex stimuli further. Moreover, if meaning can be derived from such complexity, then this can give rise to increased interest as well as elevated liking that contrasts with an initial negative appraisal at the default processing stage.

Experiment 1 only provided participants with a fixed and relatively short viewing time of 8 s for each presented image. This standardized viewing time has advantages in terms of controlling for the exposure duration of images and thereby mitigating methodological difficulties with time-locking the EEG recordings to the presentation of stimuli, which would arise from giving people an unconstrained viewing time. That said, one key disadvantage with an 8 s viewing period is that this might limit people’s opportunity to engage more fully in perceiver-driven elaborative reasoning processes, which might be prematurely curtailed when the 8 s viewing window terminates. Such curtailment of reflective processing might weaken the emergence of the phenomenological and/or psychophysiological correlates of aesthetic appraisals arising at the reflective processing stage. Admittedly, the very compelling phenomenological evidence deriving from Experiment 1 suggests that participants were indeed able to engage in reflective processing, given the marked differences in subjective rating data across the LCLF, MCMF and HCHF image categories that were fully in line with predictions. Still, the EEG data in Experiment 1 were arguably more limited in informing an understanding of differences in aesthetic experiences across conditions, and additional image-viewing time might give rise to richer psychophysiological data.

Extending the viewing time for all images also affords an opportunity to partition the viewing time so as to acquire an initial subjective measure of a participant’s liking for an image that is then followed by a second measure of liking after a further period of reflection. This “dual-response paradigm” has featured extensively in reasoning research over the past decade or so and has been highly informative for theoretical advancement relating to the nature of intuitive and reflective reasoning processes (e.g., see [90]; [91]). In the experiment that we report below, we implemented a 16 s viewing period for each presented image, with the first 6 s representing the initial default response window and the subsequent 10 s representing the subsequent reflective response window.

Only aesthetic liking ratings, complexity ratings and conceptual fluency ratings were elicited from participants for the 6 s viewing time, with the full set of phenomenological ratings as in Experiment 1 only being requested at the end of the 16 s viewing session. Methodologically, a 16 s viewing time, and its respective initial and reflective time divisions, was established based on the findings from the pre-categorization study, which revealed that in 95% of the trials the participants spent a minimum of 5 s and a maximum of 15 s on any given image. Theoretically, this experimental manipulation also aligns with evidence that complex visual stimuli take longer to process than simpler visual stimuli ([12]; [72]; [82]; [102]).

Overall, the aim of Experiment 2 was to examine whether the deployment of a two-response paradigm would reveal changes in liking judgments between the intuitive and reflective stages for images in the high-interest category (HCHF)—and potentially also in the moderate-interest category (MCMF)—relative to the low-interest category (LCLF), in line with what might be expected according to the PIA Model, whereby reflective elaboration time is needed to move from initial negative appraisals to subsequent positive appraisals. Experiment 2 also provided an opportunity to explore whether all of the other subjective measures from Experiment 1 were stable over longer viewing periods for images. Finally, the experiment provided a means to examine further the EEG correlates of aesthetic experiences over a longer time period, and more specifically whether images of a high interest level (HCHF) would continue to elicit increased alpha power relative to images in the other categories, as seen in Experiment 1.

### 3.1. Method

#### 3.1.1. Design

This study involved a 3 × 2 repeated-measures design, with one independent variable being the level of interest of the presented images (i.e., image categories LCLF, MCMF and HCHF)—as in Experiment 1—and the other independent variable being the time of the viewing, with two levels: either the initial viewing time (the first 6 s) or the reflective viewing time (after a further 10 s).

#### 3.1.2. Participants

The sample (*N* = 16) involved 8 males and 8 females aged between 18 and 45 years old (*M* = 27.00, *SD* = 7.40), none of whom had participated in Experiment 1 or the pre-categorization study. The recruitment method, inclusion and exclusion criteria and remuneration rate were consistent with those reported for Experiment 1.

#### 3.1.3. Materials and Procedure

The same stimulus materials (i.e., street-art images) used in Experiment 1 were used in the present experiment and all data-collection procedures remained the same as well, except for participants being asked to consider their initial impression of aesthetic liking, complexity and conceptual fluency after the first 6 s of image viewing. They were informed that after doing this they would have a further 10 s of image-viewing time to reflect on their first impressions of the image. All rating scales used in Experiment 2 were identical to those used in Experiment 1, with ratings relating to aesthetic liking, complexity, conceptual fluency, overall perception of flow, concentration, time distortion, arousal and pleasantness. Alpha power was also measured using the same method as in Experiment 1.

### 3.2. Results

As with Experiment 1, we first present the results relating to participants’ phenomenological ratings before we report the EEG findings. All subjective data were analyzed using repeated-measures analysis of variance (ANOVA), with Bonferroni adjustments applied for all post hoc comparisons.

#### 3.2.1. Aesthetic Liking, Complexity and Conceptual Fluency

In line with Experiment 1, ANOVA revealed that participants’ subjective ratings in relation to aesthetic liking (see Figure 3) were significantly different across the interest levels of the image categories, *F*(1.70, 242.97) = 32.96, *p* < .001, *η_p_*^2^ = 0.19, with a linear pattern of increased liking from LCLF images through to HCHF images. The main effect of time of viewing was also significant, *F*(1.00, 143.00) = 38.67, *p* < .001, *η_p_*^2^ = 0.21, with aesthetic liking being rated as higher after a period of reflection as opposed to after the initial response. The interaction effect between time of viewing and level of interest, however, was not significant, *F*(1.22, 173.77) = 0.152, *p* = .747, *η_p_*^2^ = 0.00, indicating that increased viewing time had a uniformly positive influence in increasing aesthetic liking irrespective of the nature of the images being looked at.

In relation to conceptual fluency, ANOVA revealed that participants’ ratings (see Figure 4) were significantly different across the interest levels of the image categories, *F*(1.85, 264.35) = 78.36, *p* < .001, *η_p_*^2^ = 0.35, with a linear pattern of increased perceptions of conceptual fluency from LCLF images through to HCHF images. This result supports the success of the conceptual fluency manipulation. The main effect of time of viewing was also significant, *F*(1.00, 143.00) = 146.47, *p* < .001, *η_p_*^2^ = 0.51, with the conceptual fluency of images being rated as higher after a period of reflection compared to after the initial response. The interaction between time of viewing and level of interest was also significant, *F*(2, 286) = 12.47, *p* = < .001, *η_p_*^2^ = 0.08, with time of viewing having an increasing impact on conceptual fluency ratings across the three increasing levels of interest for the images, with the greatest impact arising for images in the HCHF category.

In terms of complexity, ANOVA indicated that participants’ ratings (see Figure 4) were significantly different across the interest levels of the image categories, *F*(1.87, 266.75) = 79.06, *p* < .001, *η_p_*^2^ = 0.36, with a linear pattern of increased perceptions of complexity from LCLF images through to HCHF images. This result supports the success of the complexity manipulation. The main effect of time of viewing was also significant, *F*(1.00, 143.00) = 80.29, *p* < .001, *η_p_*^2^ = 0.36, with image complexity being rated as higher after a period of reflection compared to after the initial response. The interaction between time of viewing and image category was also significant, *F*(2, 286) = 5.38, *p* < .005, *η_p_*^2^ = 0.04, with time of viewing having an increasing impact on complexity ratings across the three levels of interest for the images, with the greatest impact arising for images in the HCHF category.

#### 3.2.2. Flow, Concentration, Time Distortion, Arousal and Pleasantness

Table 2 shows means, standard deviations, *F*-values, *p*-values and effect-size measures for the phenomenological data relating to the overall perception of flow, concentration, time distortion, arousal and pleasantness that were elicited at the end of the 16 s viewing session for each image.

A repeated-measures ANOVA indicated that participants’ overall perception of flow was significantly different across levels of interest for the image categories, *F*(2, 286) = 45.19, *p* < .001; *η_p_*^2^ = 0.20, with follow-up tests showing higher ratings for HCHF images than for either MCMF images (*d* = 0.18) or LCLF images (*d* = 0.68), as well as higher ratings for MCMF images than for LCLF images (*d* = 0.48). This result supports the same effect observed in Experiment 1. In addition, the measure of participants’ self-rated concentration showed similar effects to those seen in Experiment 1. Self-rated concentration was significantly different across levels of interest for the image categories, *F*(2, 286) = 34.99, *p* < .001; *η_p_*^2^ = 0.20, with follow-up tests showing higher ratings for HCHF images than for either MCMF images (*d* = 0.10) or LCLF images (*d* = 0.67), as well as higher ratings for MCMF images than for LCLF images (*d* = 0.60). Likewise, the analysis of participants’ self-rated time distortion was significantly different across levels of interest for the image categories, *F*(2, 286) = 29.10, *p* < .001; *η_p_*^2^ = 0.17, with follow-up tests showing higher ratings for HCHF images than for either MCMF images (*d* = 0.29) or LCLF images (*d* = 0.62), as well as higher ratings for MCMF images than LCLF images (*d* = 0.33).

Experiment 2 also produced similar findings to Experiment 1 in relation to people’s arousal and pleasantness ratings in response to the different levels of interest of the presented images. For reported feelings of arousal, a repeated-measures ANOVA showed that participants’ sense of activation was significantly different across levels of interest for the image categories, *F*(2, 286) = 20.31, *p* < .001; *η_p_*^2^ = 0.12, with follow-up tests indicating higher ratings for HCHF images than for either MCMF images (*d* = 0.08) or LCLF images (*d* = 0.53), as well as higher ratings for MCMF images than LCLF images (*d* = 0.48). For reported feelings of pleasantness, a repeated-measures ANOVA showed that participants’ feeling of pleasantness was significantly different across levels of interest for the image categories, *F*(2, 286) = 52.47, *p* < .001; *η_p_*^2^ = 0.27, with follow-up tests indicating higher ratings for HCHF images than for either MCMF images (*d* = 0.54) or LCLF images (*d* = 1.13), as well as higher ratings for MCMF images than for LCLF images (*d* = 0.56).

#### 3.2.3. EEG Findings Relating to Alpha Power

All EEG data were handled in the same way as in Experiment 1 and were likewise analyzed using repeated-measures ANOVAs, with Bonferroni adjustments applied for all post hoc comparisons. We note in advance that although alpha power was sampled at 19 key electrode sites, we only report in this section the findings from four regions (T3, T5, O1 and O2), as these were the only regions to reveal significant differences across the image categories. Topographical heat maps were created based on the absolute power values for the alpha frequency band and are presented in Figure 5. Examination of these heat maps indicates various patterns of alpha-wave activity in response to LCLF, MCMF and HCHF image categories, as well as across the initial and reflective image-viewing times.

Alpha power showed a significant main effect across image categories (HCHF, MCMF, LCLF) at T3 (*p* < .005), at T5 (*p* < .01), at O1 (*p* < .001) and at O2 (*p* < .005). No other significant effects were found. More specifically, at T3 and T5, alpha power showed a significant decreasing linear trend across the three image categories, whereas at O1 and O2, alpha power showed an increasing linear trend across the image categories (see Figure 6).

Additionally, there was a significant main effect of the time-of-viewing factor in all channels except for Fp1, F7, F8, T4, Pz and O2, with most *p* values < .001. This suggests that there are differences in the neural patterns associated with initial versus reflective aesthetic appraisals of graffiti street art that implicate the involvement of multiple brain regions during aesthetic appraisals.

### 3.3. Discussion

A key aim of Experiment 2 was to test the dual-process assumptions of [42] ([42]) PIA Model, which proposes that people first engage in an initial automatic, *percept-driven* process that produces a default judgment of liking and then engage in a subsequent *perceiver-driven* reflective process that leads to a final judgment of liking. Stimulus complexity, moreover, is viewed as a pertinent trigger for people to engage in the second, reflective stage, as complexity can serve to motivate people to apply elaborative reasoning and to pursue the derivation of meaning. Such reflective processing can thence give rise to increased interest as well as elevated liking. By partitioning viewing times for all images into a short initial stage (6 s) followed by a longer (10 s) stage aimed at facilitating reflection, we predicted that phenomenological judgments relating to aesthetic liking might show a more marked increase for images in the high-interest-level category (HCHF)—and also potentially in the moderate-interest category (MCMF)—relative to the low-interest category (LCLF).

In contrast to our predictions, however, the interaction effect between time of viewing and level of interest in relation to liking ratings was not significant. Instead, the main effect of time of viewing was significant, with additional processing time appearing to enhance aesthetic appraisals for *all* image categories. Experiment 2 also replicated the linear pattern of increased liking from LCLF images through to HCHF images, as seen in Experiment 1, suggesting that the effect of level of interest on aesthetic liking remains robust and stable across longer viewing times (i.e., 16 s in Experiment 2 vs. 8 s in Experiment 1). Regarding the absence of a predicted interaction between time of viewing and level of interest, we admit this presents something of an explanatory challenge—as does the observation that increased viewing time generally leads to more positive aesthetic appraisals. One possibility is that artificially *imposing* additional viewing time on participants—including for images in the LCLF category—serves to encourage greater elaboration and search after meaning, which increases aesthetic liking across all image categories.

In this latter respect, a better test of [42] ([42]) PIA Model using a dual-response paradigm would be to allow participants to provide ratings whenever they wish to during the second image-viewing stage. Permitting participants to terminate their image viewing whenever they decide to do so would allow them to disengage early from images in the LCLF category, which are both simple and conceptually disfluent, whilst allowing them to engage longer and more productively with images in the HCHF category, triggered by image complexity and emerging interest as meaning is manifested. In this way, the predicted interaction effect between time of viewing and level of interest in relation to liking ratings should be discernable in the data. We suggest that such a study represents a worthwhile future line of experimentation.

Before moving on from considering the phenomenological data derived in Experiment 2, we note another possible reason for the absence of a predicted interaction between time of viewing and level of interest in relation to subjective judgments of liking. We suggest that the mere act of eliciting aesthetic ratings from participants at the initial 6 s time-point could have biased subsequent evaluations, thereby leading to uniform increases in aesthetic evaluations at the second time-point. That is, merely taking phenomenological measurements at the first time-point could have primed participants’ thoughts and feeling during the second viewing stage, essentially focusing their attention on aesthetic aspects of the images that had already been probed and evaluated. This explanation would be valuable to explore further in the context of studying aesthetic liking, not least because it raises serious questions regarding the viability of deploying a two-response paradigm to study changes in aesthetic judgments over time.

In terms of the EEG data, the findings relating to alpha power were far from straightforward. Given the absence of a predicted interaction between time of viewing and level of interest in the phenomenological data relating to aesthetic liking, it was unsurprising that the EEG analysis of alpha power likewise revealed the absence of such an interaction effect. This result could potentially be explained in the same ways as discussed above in relation to the phenomenological data. A main effect of the level of interest for the image categories was, however, seen at four electrode sites (that is, T3, T5, O1 and O2), with the temporal brain regions (T3 and T5) showing a significantly decreasing linear trend in alpha power across the three image categories and the occipital regions (O1 and O2) showing a significantly increasing linear trend in alpha power across the image categories.

Although the T5 trend was like the one observed in Experiment 1, we note that the T3 trend was the reverse of what was found in Experiment 1. This contradictory finding is difficult to explain given that it arose even in the initial 6 s image-viewing window in Experiment 2 relative to the very similar 8 s viewing window in Experiment 1. The best explanation that we can offer for the contradictory findings relates to instructional changes that were implemented across the two experiments. More specifically, participants in Experiment 2 knew in advance that for each presented image they would be asked to consider their initial impression of aesthetic liking, complexity and conceptual fluency after the first 6 s of image viewing and that they would have a further 10 s of image-viewing time to reflect on their first impressions of the image. It could well be that the effect of these instructions was to induce greater visual attention (underpinned by activation at the T3 electrode site) toward lower-interest images than higher-interest ones because of their lack of conceptual fluency, perhaps driven by the knowledge that extensive processing time was available.

Regarding the linear trend observed at O1 and O2 for increased alpha power across the interest level of images, we note that such an effect was absent in Experiment 1 and was, therefore, again possibly associated with the instructional changes across experiments, including the up-front knowledge in Experiment 2 that substantial processing time would be available for participants to attend carefully and systematically to presented stimuli so as to derive meaning and understanding from them. Increased alpha power at O2 typically reflects the recruitment of the ventral attention network to reduce distraction and enhance selective attention ([54]; [56]; [57]; [69]). Likewise, [105] ([105]) found that alpha activity in the occipital lobe is related to the sensory gating of information from the visual cortex to the ventral attention network, which leads to selective attention during stimulus processing. Owing to the complex, detailed and meaningful nature of the HCHF images, we suggest that participants may have been attempting to reduce visual distractions so that they could process the images more effectively from the outset.

## 4. General Discussion

The present study represents part of an ongoing movement in research on empirical aesthetics that aims to depart from traditional processing-fluency accounts of aesthetic liking ([82]; [102]) and instead to develop more sophisticated theories that are better able to explain a wider range of often rather nuanced findings. Such findings include those relating to the surprising way in which highly complex stimuli can often be viewed as pleasurable (e.g., [7]; [55]; [66]; [71]; [73]). We suggest that one theory that is central to contemporary conceptual advancement is [42] ([42]) Pleasure-Interest Model of Aesthetic Liking (PIA Model). This model proposes that aesthetic preferences arise from two fluency-based processes: (1) an initial automatic, percept-driven default process; (2) a subsequent perceiver-driven reflective process, which can override judgments arising at the default processing stage. Furthermore, one stimulus cue that has been mooted as being critical for catalysing further reflective engagement is that of stimulus complexity. Importantly too, if meaning can be derived from such complexity, then this can engender increased interest and elevated liking, thereby explaining why complex stimuli may have aesthetic appeal.

In previous research, [8] ([8]) tested key assumptions of the PIA Model in an experiment that simultaneously manipulated the complexity (low vs. high) of presented stimuli (abstract artworks) and their conceptual fluency (across five linearly increasing levels). [8] ([8]) found good evidence to support the PIA Model in terms of the emergence of a predicted interaction between stimulus complexity and conceptual fluency, with findings indicating that complexity modulated the effect of conceptual fluency in relation to positive aesthetic appraisals. More specifically, significant differences in beauty ratings between high- and low-complexity artworks only arose across the three highest levels of conceptual fluency and not in beauty ratings of high- and low-complexity artworks at the two lowest levels of conceptual fluency. These results clarify that people like more complex visual stimuli compared to simpler ones, but only if they can readily derive meaning from them.

The present research aimed to provide a conceptual replication of aspects of [8] ([8]) study to provide further evidence in support of [42] ([42]) PIA Model. More specifically, we established three categories of images (graffiti street art) by systematically combining complexity with conceptual fluency. These image categories spanned three levels of interest ranging from low to high: low interest (i.e., low complexity and low conceptual fluency (LCLF)), moderate interest (i.e., moderate complexity and moderate conceptual fluency (MCMF)) and high interest (high complexity and high conceptual fluency (HCHF)). Our primary behavioral prediction was that people’s phenomenological ratings of beauty should increase linearly across these three levels of interest. Both Experiments 1 and 2 showed this predicted effect, which was robust against changes in procedures across experiments, including increased image-viewing times in Experiment 2. Our secondary behavioral prediction was that people’s experience of engaging in increasingly positive aesthetic appraisal across the three image categories (LCLF, MCMF and HCHF) should be associated phenomenologically with being in a flow-like state ([30]; [31]). Again, this prediction was upheld in both Experiments 1 and 2, which showed linear increases in measures of flow (including concentration and time distortion), as well as in arousal and pleasantness, as the interest level of presented images increased.

Experiment 2 was also designed to explore whether differences in liking ratings across levels of interest for image categories would arise when an initial viewing stage (6 s window) was contrasted with a subsequent viewing window (an additional 10 s). The rationale for this “two-response” manipulation was to investigate whether initial default liking judgments would be overridden when permitted additional viewing time, especially for the image categories at higher levels of interest. The predicted interaction between time of viewing and level of interest did not emerge in the data on aesthetic liking. Intriguingly, a main effect of viewing time was instead found for all image categories, indicating that additional viewing time enhanced aesthetic liking irrespective of image complexity or conceptual fluency.

We suggested above two possible explanations for the absence of a predicted interaction between time of viewing and level of interest on judgments of aesthetic liking, with both explanations being related to methodological changes between Experiment 1 and Experiment 2 that are worthy of further investigation. First, the finding might be an artefact of essentially *enforcing* participants to engage in additional processing of all presented images (including simple but conceptually disfluent ones). It could well be the case that *self-paced* image viewing during the second, reflective stage would reveal a predicted interaction effect. Second, the finding might have arisen because eliciting aesthetic ratings from participants at the initial 6 s time-point could have primed—and thereby biased—their subsequent evaluations, thereby leading to uniform increases in these evaluations at the second time-point. Notwithstanding the absence of the predicted interaction between time of viewing and level of interest on liking judgments in Experiment 2, we nevertheless contend that our phenomenological data across both experiments provide good support for the predictions of the PIA Model and corroborate the importance of elaborative engagement in driving aesthetic liking for images with higher interest value, as captured by their complexity and conceptual fluency.

Experiments 1 and 2 were not only designed with the aim of testing predictions relating to the phenomenology of aesthetic liking and flow experiences but also to examine the psychophysiological correlates of such phenomenological states. To this end, cortical EEG measures were taken across 19 electrode sites during image viewing in both experiments. The data analysis focused on alpha activity, as elevations in alpha power have been linked to aesthetic judgments of beauty, including aesthetic appraisals of visual artworks ([25], [26]). The EEG data revealed some important findings regarding cortical regions that appear to be associated with positive aesthetic appraisals. Considering the two experiments in aggregate, significant changes in alpha power across image categories (LCLF, MCMF and HCHF) were associated with temporal regions (T3, T5 and T6) and occipital regions (O1 and O2).

The T3 electrode site is related to the ventral attention network, including visual perception and encoding processes ([59]), and likewise the T6 electrode site has been linked to object recognition and visual memory ([17]; [46]). The finding in Experiment 1 that increased alpha power arose in these temporal regions when participants were viewing HCHF images relative to other images is suggestive of greater visual engagement with the former, including recruiting resources related to visual recollection during the meaningful interpretation of such high-complexity but conceptually meaningful images. We note, however, that Experiment 2 revealed no significant differences in alpha power across image categories at T6, and the opposite trend was observed in relation to the T3 electrode to that seen in Experiment 1. We propose that this oppositional effect might again be attributable to methodological changes between Experiment 1 and Experiment 2. In particular, participants in Experiment 2 had prior knowledge from the given instructions that for each presented image they would be asked to consider their initial impression of aesthetic liking, complexity and conceptual fluency after an initial image-viewing period and that they would then have additional time to consider the image and provide revised judgments. We have suggested above that the effect of these instructions might have been to induce greater visual attention (reflected in cortical activation at the T3 electrode site) toward lower-interest images than higher-interest ones because of their lack of conceptual fluency—and driven by the participants’ knowledge that extensive processing time was available. We acknowledge the speculative nature of this explanation, which points to the need for replication studies to determine the reliability of the observed differential T3 effect in Experiment 1 and Experiment 2.

Experiment 1 also showed an effect of increased alpha power at electrode site T5 when participants were viewing images in the LCLF category relative to the HCHF category. This same effect was also observed in Experiment 2. T5 is linked to the transference of visual information into semantic categories via language processing (e.g., [32]; [48]; [47]), suggesting that viewing images of low conceptual fluency may have required the use of more resources to place conceptually challenging image features into semantic categories.

The alpha power differences arising in occipital regions (O1 and O2) were only observed in Experiment 2, which involved longer overall viewing times than Experiment 1, and indicated increased alpha power in these regions across increasing levels of interest for the image categories. As we noted previously, greater alpha power at O2 often reflects the recruitment of the ventral attention network to reduce visual distraction through processes involving idling and inhibition ([54]; [56]; [57]; [69]), with the occipital lobe generally being implicated in selective attention during stimulus processing (e.g., [105]). It is, therefore, perhaps not surprising that highly complex and meaningful visual images will necessitate interpretation through a reduction in distractions.

### Limitations and Future Research

Some aspects of our phenomenological findings from Experiment 2, as well as the associated psychophysiological data, are far from straightforward to interpret. With the benefit of hindsight, we believe that artificially imposing additional viewing time on participants in this experiment—including for images of low interest value (i.e., those in the LCLF category that were simplistic but also seemingly devoid of meaning)—may have inadvertently encouraged participants to pursue greater elaboration and search after meaning, thereby increasing aesthetic liking across all image categories over time and complicating data interpretation.

As we have suggested, a better test of [42] ([42]) PIA Model using a dual-response paradigm would be to allow participants to self-pace during the second image-viewing stage and provide subjective ratings at whatever point they feel is appropriate. Permitting participants to self-terminate their image viewing would allow for a much more natural test of the predictions of the PIA Model and is an important direction for future experimentation. It is also likely that the EEG data relating to alpha power changes over time would be more informative when implementing such a self-paced two-response paradigm.

We finally note that previous studies examining EEG markers of aesthetic processing have not only examined alpha waves but also other brain waves, such as beta, theta, delta and gamma (e.g., [20]; [25]; [56]; [63]). As such, it would be valuable for future studies examining the psychophysiological markers of aesthetic liking and flow experiences with complex yet meaningful stimuli to broaden data analysis to include other EEG brain waves, albeit in a theoretically informed manner.

Further suggestions for future research include taking a multi-method approach to examining aesthetic responses to images of varying complexity and conceptual fluency so as to build up a rich picture of evidence for theories such as the PIA Model. For example, facial expressions could be investigated using electromyography to measure the initial and reflective impressions that people have of images of different complexity and conceptual fluency levels (cf. [33]; [35]; [36]; [41]). Measuring electromyography activity would usefully supplement self-report data relating to phenomenological experiences of arousal and pleasantness. Furthermore, research has examined aesthetic appraisals by including eye tracking to examine gaze patterns ([74]), which we contend could be very useful in future research investigating aesthetic processing and attention allocation during the viewing of artworks of different levels of complexity and conceptual fluency.

## Figures and Tables

**Figure 1 jintelligence-12-00042-f001:**
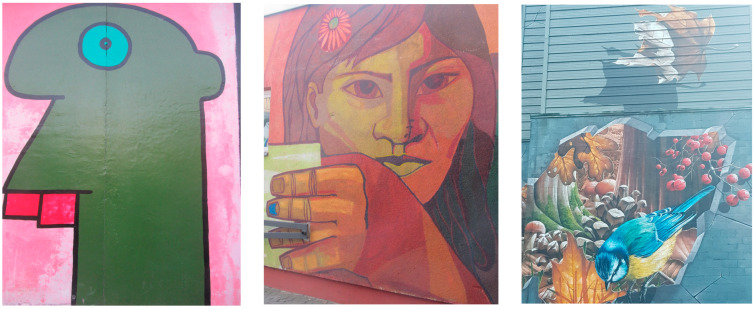
Example images selected from image categories across the three increasing levels of interest. From left to right: low complexity and low conceptual fluency (LCLF); moderate complexity and moderate conceptual fluency (MCMF); and high complexity and high conceptual fluency (HCHF).

**Figure 2 jintelligence-12-00042-f002:**
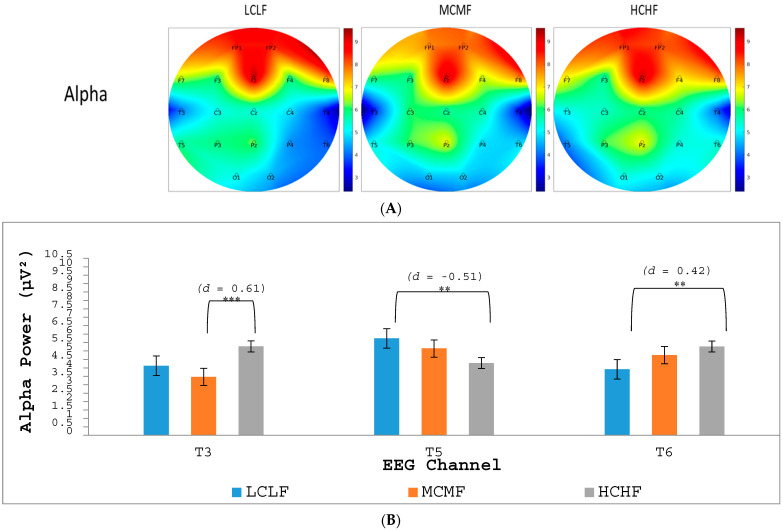
Topographical heat maps for absolute alpha power across all 19 EEG channels measured as µV^2^ and ranging from lowest (in blue) to highest (in red) during the viewing of images at different levels of interest: LCLF, MCMF and HCHF (**A**). Significant differences in absolute alpha power across EEG channels during the viewing of images at different levels of interest. ** *p* < .005; *** *p* < .001 (**B**).

**Figure 3 jintelligence-12-00042-f003:**
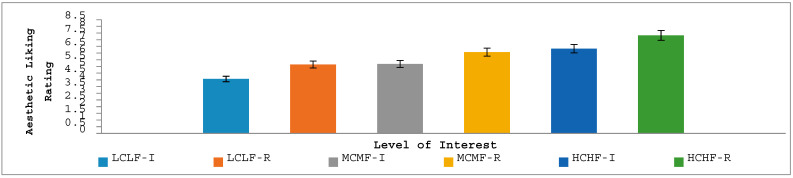
Mean aesthetic liking ratings for images at different levels of interest (LCLF, MCMF, HCHF) as a function of time of viewing (I = immediate; R = reflective). Errors bars are standard errors of the mean.

**Figure 4 jintelligence-12-00042-f004:**
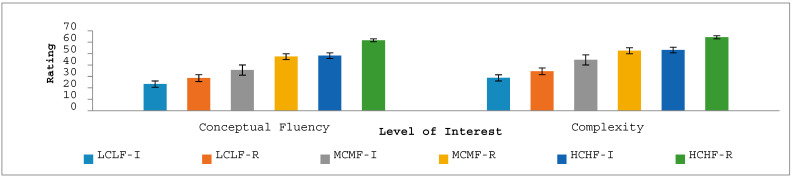
Mean ratings of conceptual fluency and complexity for images across different levels of interest (LCLF, MCMF, HCHF) as a function of time of viewing (I = immediate; R = reflective). Errors bars are standard errors of the mean.

**Figure 5 jintelligence-12-00042-f005:**
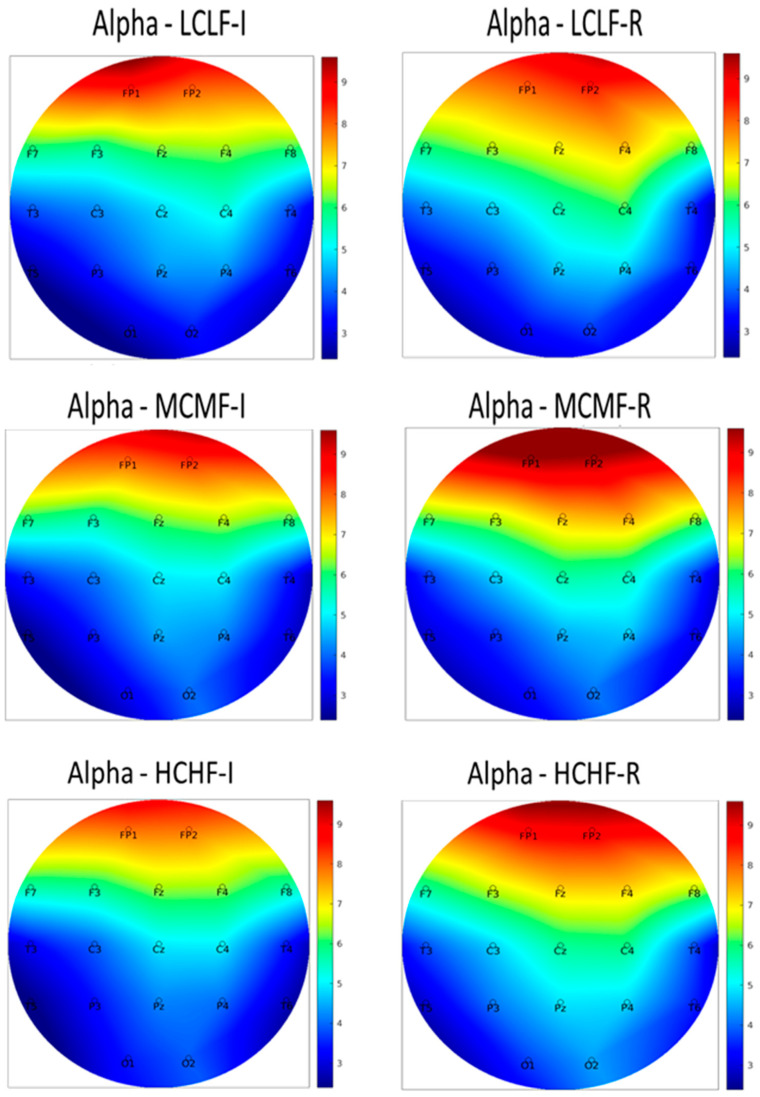
Topographical heat maps for absolute alpha power across all 19 EEG channels, measured as µV^2^ and ranging from lowest (in blue) to highest (in red) during the viewing of images at different levels of interest and at different times of viewing (I = immediate; R = reflective): LCLF-I (**top left**), LCLF-R (**top right**), MCMF-I (**middle left**), MCMF-R (**middle right**), HCHF-I (**bottom left**) and HCHF-R (**bottom right**).

**Figure 6 jintelligence-12-00042-f006:**
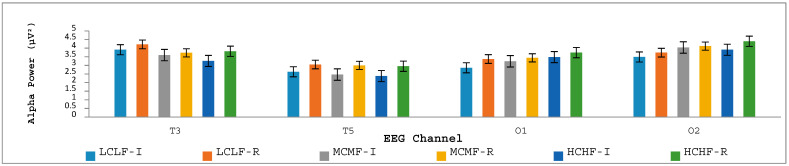
Mean alpha power across the levels of interest conditions (LCLF, MCMF, HCHF) for the four EEG channels (T3, T5, O1 and O2) that revealed a significant main effect. Note that no interaction effects were significant that involved the time-of-viewing factor (I = immediate; R = reflective).

**Table 1 jintelligence-12-00042-t001:** Means (*M*) and standard deviations (*SD*) for phenomenological rating data in Experiment 1 across levels of interest for image categories (LCLF, MCMF and HCHF) and outcomes of ANOVAs and post hoc *t*-tests, including *p*-values and measures of effect sizes.

Dependent Variables	LCLF *M* (*SD*)	MCMF *M* (*SD*)	HCHF *M* (*SD*)	*F*(2, 286) *(η_p_*^2^)	Post Hoc Comparison (*p*-Value and *d*) LCLF vs. MCMF	Post Hoc Comparison (*p*-Value and *d*) MCMF vs. HCHF	Post Hoc Comparison (*p*-Value and *d*) LCLF vs. HCHF
Aesthetic liking	4.03 (2.73)	5.60 (2.60)	6.71 (2.42)	40.50 *** (0.22)	LCLF < MCMF (*p* < .001, *d* = 0.36)	MCMF < HCHF (*p* = .001, *d* = 0.44)	LCLF < HCHF (*p* < .001, *d* = 1.04)
Complexity	30.76 (22.50)	47.43 (24.46)	60.69 (25.77)	54.21 *** (0.28)	LCLF < MCMF (*p* < .001, *d* = 0.71)	MCMF < HCHF (*p* < .001, *d* = 0.53)	LCLF < HCHF (*p* < .001, *d* = 1.24)
Conceptual fluency	25.63 (22.24)	38.40 (24.80)	52.01 (29.46)	50.09 *** (0.26)	LCLF < MCMF (*p* < .001, *d* = 0.54)	MCMF < HCHF (*p* < .001, *d* = 0.50)	LCLF < HCHF (*p* < .001, *d* = 1.01)
Overall perception of flow	4.66 (2.75)	5.94 (2.16)	6.56 (1.87)	36.80 *** (0.21)	LCLF < MCMF (*p* < .001, *d* = 0.52)	MCMF < HCHF (*p* = .005, *d* = 0.81)	LCLF < HCHF (*p* < .001, *d* = 0.81)
Concentration	6.25 (2.63)	6.56 (2.18)	7.22 (2.01)	11.33 *** (0.07)	LCLF < MCMF (*p* = .525, *d* = 0.13)	MCMF < HCHF (*p* < .001, *d* = 0.31)	LCLF < HCHF (*p* < .001, *d* = 0.41)
Time distortion	4.24 (2.63)	4.68 (2.81)	5.42 (2.59)	9.30 *** (0.06)	LCLF < MCMF (*p* = .191, *d* = 0.16)	MCMF < HCHF (*p* = .044, *d* = 0.27)	LCLF < HCHF (*p* < .001, *d* = 0.45
Arousal	5.51 (2.35)	5.86 (2.16)	6.52 (1.97)	15.81 *** (0.21)	LCLF < MCMF (*p* = .195, *d* = 0.16)	MCMF < HCHF (*p* = .001, *d* = 0.32)	LCLF < HCHF (*p* < .001, *d* = 0.47)
Pleasantness	4.88 (2.44)	5.69 (2.18)	6.63 (2.08)	25.92 *** (0.21)	LCLF < MCMF (*p* = .003, *d* = 0.35)	MCMF < HCHF (*p* < .001, *d* = 0.44)	LCLF < HCHF (*p* < .001, *d* = 0.77)

Note. LCLF = low complexity, low conceptual fluency; MCMF = moderate complexity, moderate conceptual fluency; HCHF = high complexity, high conceptual fluency. *** *p* < .001.

**Table 2 jintelligence-12-00042-t002:** Means (*M*) and standard deviations (*SD*) for phenomenological rating data (taken after the reflective viewing stage) in Experiment 2 across levels of interest for image categories (LCLF, MCMF and HCHF), as well as outcomes of ANOVAs and post hoc *t*-tests, including *p*-values and measures of effect sizes.

Dependent Variables	LCLF *M* (*SD*)	MCMF *M* (*SD*)	HCHF *M* (*SD*)	*F*(2, 286) *(η_p_*^2^)	Post Hoc Comparison (*p*-Value and *d*) LCLF vs. MCMF	Post Hoc Comparison (*p*-Value and *d*) MCMF vs. HCHF	Post Hoc Comparison (*p*-Value and *d*) LCLF vs. HCHF
Overall perception of flow	5.08 (2.22)	6.19 (2.41)	6.60 (2.25)	45.19 *** (0.20)	LCLF < MCMF (*p* < .001, *d* = 0.48)	MCMF < HCHF (*p* = .053, *d* = 0.18)	LCLF < HCHF (*p* < .001, *d* = 0.68)
Concentration	6.28 (1.92)	7.38 (1.76)	7.56 (1.91)	34.99 *** (0.21)	LCLF < MCMF (*p* < .001, *d* = 0.60)	MCMF < HCHF (*p* = .695, *d* = 0.10)	LCLF < HCHF (*p* < .001, *d* = 0.67)
Time distortion	4.47 (2.83)	5.41 (2.91)	6.25 (2.91)	29.10 *** (0.17)	LCLF < MCMF (*p* < .001, *d* = 0.33)	MCMF < HCHF (*p* = .001, *d* = 0.29)	LCLF < HCHF (*p* < .001, *d* = 0.62)
Arousal	5.60 (1.89)	6.43 (1.53)	6.57 (1.79)	20.31 *** (0.12)	LCLF < MCMF (*p* < .001, *d* = 0.48)	MCMF < HCHF (*p* = .999, *d* = 0.08)	LCLF < HCHF (*p* < .001, *d* = 0.53)
Pleasantness	4.72 (2.13)	5.88 (2.01)	6.87 (1.63)	52.47 *** (0.27)	LCLF < MCMF (*p* < .001, *d* = 0.56)	MCMF < HCHF (*p* < .001, *d* = 0.54)	LCLF < HCHF (*p* < .001, *d* = 1.13)

Note. LCLF = low complexity, low conceptual fluency; MCMF = moderate complexity, moderate conceptual fluency; HCHF = high complexity, high conceptual fluency. *** *p* < .001.

## Data Availability

The data that support the findings of this study are available at https://osf.io/vczwd/, accessed on 11 March 2024.

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
