# Peer review of "Stimulus Complexity Can Enhance Art Appreciation: Phenomenological and Psychophysiological Evidence for the Pleasure-Interest Model of Aesthetic Liking"

_jintelligence, 2024, doi:10.3390/jintelligence12040042_

Round 1

Reviewer 1 Report

Comments and Suggestions for Authors

            This paper sought to study the experiential and physiological consequences of viewing artworks with varying degrees of visual and conceptual complexity. In line with the predictions of the Pleasure-Interest Model of Aesthetic Liking (PIA) and the author’s previous work, the two experiments of this article found that as the level of complexity (or interest in the authors’ terms) increased, measures of participant’s subjective experience such as aesthetic liking and perception of being in a flow state also increased. In contrast, the EEG findings appear to contradict themselves across the two studies, which the authors argue is indicative of a methodological constraint in the current studies (i.e., participants were forced to view the artworks for a set period of time).

            Overall, I found this to be an intriguing and worthwhile contribution to the literature, and the argument that was advanced was compelling and relatively complete. With that being said, I had some concerns that I believe should be addressed before this paper is accepted for publication. I list these concerns below:

1.      On page 5, starting at line 242, the authors discuss their prior work on this subject (i.e., Ball et al., 2018) and note that they found that stimulus complexity and conceptual complexity interact in predicting measures of aesthetic experience. Indeed, Ball et al. (2018) tested this interaction with a split between low and high visual complexity and between 5 levels of conceptual complexity and found a significant interaction between these two variables. Because of this, I was surprised to find that in the current study, the authors opted to collapse these two variables into a single variable that does not allow them to model this interaction appropriately. I would like to see more explanation on page 6 of why the authors did not instead create something like a 3x3 set of materials to cover low, medium, and high levels of their visual and conceptual complexity to account for this known interaction.

2.      On page 6, starting at line 281, the authors talk about flow theory and its relation to the predictions of the PIA model and their studies. While I agree that this is an important facet of aesthetic experience to study, this paragraph makes flow feel as if it were an afterthought to the study. I would like to see a bit more discussion of the relationship between flow and the PIA model earlier in the introduction to better integrate this paragraph into the paper.

3.      On page 9, beginning at line 423, the authors discuss their measurement of the various aesthetic emotions that they are interested in study. To do this, they have used individual items from existing scales that they believe measure the constructs that they are interested in. From a psychometric standpoint, this seems problematic because although the items seem to have good face validity, there is not precedent for using these items on their own. Additionally, by not using full scales to measure their constructs, we have no means to assess the reliability and validity of the measures used. I would like to see a justification for why this decision does not harm the construct validity of the studies.

4.      On page 21, starting at line 841, and pages 22-23, starting at line 936, the authors note that they found an anomalous trend in their EEG findings between Experiments 1 and 2. The reason they argue that this occurred was due to the “enforced additional viewing time” that was present for LCLF and MCMF images. This is a rather unsatisfying explanation since Experiment 1 displayed images for 8 seconds and the initial condition in Experiment 2 displayed images for 6 seconds and still had the contradictory findings. If my reading of this is correct, I’d like to see a stronger explanation for this anomaly (or even an admission that the data may not be usable as a basis for future research). However, I am not an expert on EEG, so it is possible that I am misreading these data. If that is the case, I believe that the EEG results could be explained with a bit more detail to ensure that general audiences are not misunderstanding the findings.

5.      A relatively small issue: Tables 1 and 2 display the effect sizes for the post-hoc comparisons (which I appreciate), but not the p-values for these tests. I'd like to see an indication of which post-hoc comparisons were significant, or a note indicating that all differences are significant if that is the case.

Reviewer 2 Report

Comments and Suggestions for Authors

The paper “Stimulus Complexity Can Enhance Art Appreciation: Phenom-2 enological and Psychophysiological Evidence for the Pleasure-3 Interest Model of Aesthetic Liking” examines across two experimental studies how participants respond in terms of subjective ratings and in terms of neuronal activity to stimuli varying in complexity and conceptual fluency. Overall, the pattern of results of the first study and parts of the results of the second study are in line with the predictions of the PIA model.

In sum, the present manuscript is very well-written, the authors show a deep understanding of the literature on empirical aesthetics, and they conducted two interesting experiments which were very well executed and analyzed. I truly enjoyed reading the paper and think that the authors did an excellent job in crafting their manuscript. Especially the theory section and the derivation of the research questions are outstanding and attest to a very rich knowledge of the literature. Accordingly, I have only some minor comments that mainly focus on ways in which the reader’s understanding and the flow of reading could be improved.

1)      Pre-categorization study: I did not get what exactly a curve-fit estimation is (l. 358). From my reading, it seems to be very similar to a correlation or a linear regression because it is used to justify a composite categorization. I would suggest to either explain the curve-fit estimation or to use a standard approach such as correlation or regression. Next, I did not fully understand the z-standardization (l. 360-362). Were the ratings first aggregated per image and afterward z-standardized or were the ratings per participant z-standardized and afterwards aggregated per stimulus? Please clarify. Finally, I was a bit puzzled that the manipulation of interest had such strong effects in the later studies based on the categorization rule used. A z-value of 0.1 (or -0.1) is only one-tenth of a standard deviation. Thus, the three image categories are not that far away from each other. Please check once more whether the values are correct because I had expected that stronger manipulations would be required to elicit the observed effects. But if they are correct, I am fine. Anyways, it would probably be helpful to also report the mean z-values of the nine selected images per category to illustrate the actual strength of the manipulation.

2)      Table 1: I would suggest to also add p-values for the post hoc contrasts.

3)      Section 2.2.4. (l. 515 ff.): For me, it would have been very helpful if the section had started with a short description of the number of brain regions analyzed and a short sentence declaring that only the three significant regions are reported in this section.

4)      Table 2: Please add p-values for the post hoc contrasts. Between Arousal and Pleasantness is a horizontal line that needs to be removed.

5)      Section 3.2.3 (l. 767 ff.): Again a short intro declaring that only the four significant brain regions are reported in detail would have been helpful.

6)      Figure 5: Parts of the figure are missing.

7)      Section 3.3. (l. 795 ff.): Another explanation for the inconsistent findings regarding the subjective and neuronal responses for the second measurement point is that measuring the manipulation checks after the initial stage could have biased all evaluations and neural responses during/after the longer second stage. That is, merely measuring the manipulation checks could have altered participants thinking and feeling during the second stage by shifting attention to aspects that are only considered due to the manipulation checks.

8)      If copyright permits, it would be great if three exemplary stimuli (one per category) could be included in the main paper after the description of the pre-study.

9)      Minor points and typos:

a.       l. 94: linking needs to be liking.

b.      l. 414: that that

c.       l. 622: were were

d.      l. 683: analyses needs to be analyzed

e.       l. 782: initial versus and reflective: either versus or and needs to be removed

f.       l. 838-840: Please not only name the brain regions but also the electrode sites connected to these brain regions.

g.      l. 906: likeness probably should be liking

h.      l. 1081: This paper on “Measuring processing fluency” was published in 2018, not 2017.

Round 2

Reviewer 1 Report

Comments and Suggestions for Authors

I have no further suggestions for the authors, as they have addressed my concerns for the paper adequately.

Reviewer 2 Report

Comments and Suggestions for Authors

The authors did a great job in fixing my concerns. It was a pleasure to read the revised version of the manuscript and I have no remaining concerns or suggestions. From my perspective, this version of the mansucript is very interesting and makes a nice contribution to the literature. All the best for your future research in this area!